# Targeting Cross-Presentation as a Route to Improve the Efficiency of Peptide-Based Cancer Vaccines

**DOI:** 10.3390/cancers13246189

**Published:** 2021-12-08

**Authors:** Ben Wylie, Ferrer Ong, Hanane Belhoul-Fakir, Kristin Priebatsch, Heique Bogdawa, Anja Stirnweiss, Paul Watt, Paula Cunningham, Shane R. Stone, Jason Waithman

**Affiliations:** 1Telethon Kids Institute, The University of Western Australia, Nedlands, WA 6009, Australia; ben.wylie@telethonkids.org.au; 2PYC Therapeutics, Harry Perkins Institute, QEII Medical Centre, Nedlands, WA 6009, Australia; ferrer.ong@pyctx.com (F.O.); anja.stirnweiss@pyctx.com (A.S.); paula.cunningham@pyctx.com (P.C.); 3School of Public Health, Faculty of Health Sciences, Curtin University, Bentley, WA 6102, Australia; hanane.belhoul-fakir@gmail.com; 4Exopharm, Melbourne, VIC 3000, Australia; kmpriebatsch01@gmail.com; 5Biocurate, Parkville, VIC 3052, Australia; hbogdawa@gmail.com; 6Avicena, West Perth, WA 6005, Australia; paul.m.watt@icloud.com; 7School of Agriculture and the Environment, University of Western Australia, Nedlands, WA 6009, Australia

**Keywords:** peptide vaccine, cancer vaccine, dendritic cell, cell penetrating peptide, cross-presentation, melanoma, immunotherapy

## Abstract

**Simple Summary:**

Cancer vaccination is a potential anti-cancer therapy which may improve clinical outcomes, particularly in combination with current immunotherapies, but to date has demonstrated only limited success. Here we set out to improve the effectiveness of peptide-based cancer vaccine design by implementing a unique approach to target peptide antigens to the cross-presentation pathway within dendritic cells. We demonstrate that the addition of a short cell penetrating peptide sequence to the vaccine construct enhances the efficiency of our vaccine and improves outcomes in both a viral and a cancer model. Our findings provide a simple strategy that may enhance the effectiveness of peptide-based cancer vaccines and the priming of anti-tumor immunity.

**Abstract:**

Cross-presenting dendritic cells (DC) offer an attractive target for vaccination due to their unique ability to process exogenous antigens for presentation on MHC class I molecules. Recent reports have established that these DC express unique surface receptors and play a critical role in the initiation of anti-tumor immunity, opening the way for the development of vaccination strategies specifically targeting these cells. This study investigated whether targeting cross-presenting DC by two complementary mechanisms could improve vaccine effectiveness, in both a viral setting and in a murine melanoma model. Our novel vaccine construct contained the XCL1 ligand, to target uptake to XCR1^+^ cross-presenting DC, and a cell penetrating peptide (CPP) with endosomal escape properties, to enhance antigen delivery into the cross-presentation pathway. Using a prime-boost regimen, we demonstrated robust expansion of antigen-specific T cells following vaccination with our CPP-linked peptide vaccine and protective immunity against HSV-1 skin infection, where vaccine epitopes were natively expressed by the virus. Additionally, our novel vaccination strategy slowed tumor outgrowth in a B16 murine melanoma model, compared to adjuvant only controls, suggesting antigen-specific anti-tumor immunity was generated following vaccination. These findings suggest that novel strategies to target the antigen cross-presentation pathway in DC may be beneficial for the generation of anti-tumor immunity.

## 1. Introduction

The success of checkpoint blockade immunotherapy (CPB) [1,2,3], in the treatment of solid cancers and adoptive cell therapy (ACT) [4,5,6], for the treatment of blood cancers, as well as limited solid tumors, has renewed global interest in the power of the immune system to target tumor neoantigens [7]. Despite these advances, many cancers remain resistant to immunotherapy due to the establishment of a suppressive tumor microenvironment (TME) [8] and sub-optimal priming of the immune system [9,10]. This has led to a resurgence of interest in therapeutic cancer vaccination strategies, for their potential to generate potent anti-tumor immunity [11,12]. To date, there is no clear consensus on the optimal approach for cancer vaccine development, suggesting their therapeutic potential remains unfulfilled. Here, we investigated two complementary strategies to target the antigenic payload of a peptide-based vaccine construct into the cross-presentation pathway within a unique subset of dendritic cell (DC). Critically, we demonstrate that inclusion of a cell-penetrating peptide (CPP), capable of facilitating the endosomal escape of its peptide cargo [13], is a simple and effective strategy to increase vaccine efficiency and mediate improved viral and tumor control in preclinical models of disease.

The simplest form of cancer vaccines are multivalent peptide constructs containing a pool of known disease-specific antigens, to activate the immune system. These constructs can be packaged in nanoparticles to enhance their longevity and uptake by antigen-presenting cells [14] and delivered with adjuvants to skew the immune balance towards activation rather than tolerance [15]. In general, cancer vaccination strategies have shown limited success on their own [16,17,18], however, they may be well suited to synergise with other approved immunotherapies such as CPB [19]. The recent success and rapid manufacture of mRNA vaccines for COVID-19 is an example of the power of this approach [20]. Recently, a push to develop bioinformatic pipelines to call and identify patient-specific mutations and cancer neoantigens has made ground-breaking progress into the field of personalized cancer vaccination [21], or vaccines that include a pool of the most commonly identified cancer-specific epitopes [22]. In addition, novel strategies are being developed to drive productive anti-tumor immunity using immune-stimulating ligands such as Flt3 [23,24] and MIP3a [25], which are applicable to the vaccine setting.

DC are professional antigen-presenting cells and the key sentinels of the immune system responsible for the priming of cytotoxic CD8^+^ T cells against viral- or tumor-specific antigens [26]. This immune priming relies on the cross-presentation of exogenous antigens, whereby specific subsets of DC possessing unique biology, acquire extracellular antigens and divert them into their own MHC class I presentation pathway, usually reserved for the presentation of internal antigens [27,28]. The critical role of these DC in the initiation of anti-tumor immunity, regulation of adaptive immune and the response to immunotherapy have now been well documented [29,30,31], suggesting they are a key target for therapies aiming to induce robust anti-tumor responses. Recently, the discovery of X-C Motif Chemokine Receptor 1 (XCR1) as a specific and conserved marker of cross-presenting DC [32] has allowed for the specific targeting of these subsets as a strategy to direct productive anti-tumor immunity [33,34,35]. These studies in combination with the current advances in neo-antigen discovery [22] may drive the next wave of candidate cancer vaccines, however, more remains to be done to reach the full potential of this technology.

Cross-presentation occurs when exogenous or extracellular antigens, taken up by specific DC and retained within cytosolic endosomes, are shuttled back into the cytoplasm via transporters such as Sec61 [36], for degradation by the cellular proteasome before being transported into the endoplasmic reticulum by molecular chaperones such as TAP [37]. Alternately endocytosed proteins may be degraded within the endosome by cathepsin S [38] and the necessary antigen processing machinery recruited to facilitate peptide binding to recycled MHC class I molecules within the endosome [39]. While it appears cross-presenting DC may use both these pathways, it remains unclear to what extent each pathway contributes to the generation of immunity [40]. One potential strategy to enhance the efficiency of antigen cross-presentation is to drive the escape of antigens out of endosomes and into the cytoplasm. CPP are short, charged peptides that can be useful for carrying cargo across cellular membranes [41]. In addition, some CPP exhibit endosomal escape capabilities [42] and have been shown to effectively transport their cargo out of endosomes and into the cytoplasm [13]. The linking of such a CPP to a peptide vaccine construct may represent a simple strategy to drive the efficient entry of antigen into the cross-presentation pathway. Cross-presenting DC recruit NOX2 to endosomes via RAB27A and RAC2 [43,44] leading to the production of radical oxygen species and increased endosomal pH, limiting antigen degradation, and increasing cross-presentation [45]. Special care should be taken in the design of vaccine constructs to ensure they function correctly in this environment, in this case breaking down into constituent parts to facilitate proper endosomal escape.

In this study we apply two non-overlapping techniques to target a short peptide-based vaccine construct, containing tumor-associated antigens, specifically to the cross-presentation pathway in specific DC subsets. First, we engineered a XCR1-targeting moiety using a sequence from the murine X-C Motif Chemokine Ligand 1 (XCL1) protein. Secondly, we employed our expertise in CPP development to design a CPP containing peptide vaccine, to drive endocytosed peptide antigens out of endosomes and into the cytoplasm to access the cross-presentation pathway more efficiently. Our data demonstrate that our novel vaccination strategy induces strong proliferation of antigen specific CD8^+^ T cells, with increased efficiency compared to controls. In a Herpes Simplex Virus 1 (HSV) infection model this CD8^+^ response inhibited viral replication at the site of infection. Furthermore, when targeting B16 melanoma, engineered to express the vaccine antigens, this immune response delayed tumor outgrowth. Importantly, our data show that the simple addition of an endosomal escaping CPP to a peptide-based vaccine construct can enhance vaccine efficiency leading to the induction of robust anti-viral and anti-tumor immunity. This information may be of particular interest in the current development of preclinical peptide-based vaccines and the design of future cancer trials in combination with CPB immunotherapy.

## 2. Materials and Methods

### 2.1. Vaccine Design and Construction

The peptide vaccine was designed as a multicomponent construct comprising a recombinantly expressed protein unit, and a synthetic peptide unit. The protein unit was made up of the targeting moiety, murine XCL1 (residues 22-114, Uniprot P47993, Lymphotactin), and the protein ligation moiety, SpyCatcher (also SpyC) [46]. These units were expressed as a fusion protein with affinity (6xHis) and detection (Strep-II) epitopes for purification. The synthetic peptide unit was designed to incorporate four distinct subunits, each with a specific role. The first peptide subunit was the SpyTag (also SpyT) peptide ligation tag for covalent linkage with the SpyCatcher scaffold of the protein unit. The second subunit was a linking sequence for joining the third and fourth units to the peptide ligation tag. The third subunit was the cell-penetrating peptide (1746 or TAT). The fourth peptide subunit comprised a GAS spacer sequence, the HSV glycoprotein D (gD) epitope gD_290–302_, a GTG spacer sequence, and the HSV glycoprotein B (gB) epitope gB_498–505._

The linking sequence of the peptide subunit required redesign after results described in Figure 1. Initially this linking unit comprised two amino acids (GC); this resulted in only limited presentation of antigen on MHC class I presentation, suggesting endosomal escape was not efficient. The next iteration of this linking sequence included the addition of a nine amino acid spacer sequence, and Furin and Cathepsin-B cleavage motifs (GSGTGGSTGRKKRSV-Citrulline-GC) to facilitate early and late endosomal cleavage from the targeting moiety.

### 2.2. Peptide Synthesis

Peptides were synthesized by Fmoc chemistry, at Pepscan (Lelystad, The Netherlands) as two separate components before being linked together via disulfide coupling. All peptides were purified by HPLC, and purity confirmed >95% by LC-MS.

The first peptide component, required for linking to the targeting and ligation unit, was synthesized as a single fragment, and comprised the first two subunits, namely the SpyTag peptide ligation tag and the linker sequence. The sequences comprised SpyTag (GAHIVMVDAYKPTK) and either a short spacer with a disulfide bond, or an early endosomal Furin cleavage site RKKR, and cathepsin-B endosomal cleavage site Valine-Citrulline (GC, or GSGTGGSTGRKKRSV-Citrulline-GC). All first component peptides contained a C-terminal Cysteine for coupling, with N-termini acetylated and C-termini amidated for protection. The initial first component peptides used to generate data for Figure 1 included an N-terminal V5 detection epitope but this was excluded in later peptides.

The second component, containing the cell-penetrating peptide and peptide antigen subunits, was synthesized as a single fragment. The cell penetrating peptide subunit sequences comprised of either 1746 (PLKPKKPKTQEKKKKQPPK-PKKPKTQEKKKKQPPKPKR), TAT (GRKKRRQRRRPQ) or no CPP sequence. The antigen subunit sequence comprised a short linker (GAS), the HSV gD epitope (IPPNWHIPSIQDA), a short linker (GTG), and the HSV gB epitope (SSIEFARL). Second component peptides were synthesized either with an N-terminal mercaptopropionic acid (mcpa) to allow for ligation to first component peptides by sulfur chemistry, or acetylated for protection for non-targeted applications. All C-termini were amidated for protection. 

First and second peptides were ligated together by sulfur chemistry to form a disulfide bond between the C-terminal Cysteine of component 1, and the sulfhydryl of the C-terminal mercaptopropionic acid of component 2. Conjugated peptide sequences used in manuscript are presented below: 

Synthetic peptide unit sequences for data in Figure 1.

SpyT-1746-gDgB:Ac-GKPIPNPLLGLDSTGASAHIVMVDAYKPTKG-C(nh2)-mcpa- PLKPKKPKTQEKKKKQPPKPKKPKTQEKKKKQPPKPKRGASIPPNWHIPSIQDAGTGSSIEFARL-nh2; SpyT-TAT-gDgB:Ac-GKPIPNPLLGLDSTGASAHIVMVDAYKPTKG-C(nh2)-mcpa-GRKKRRQRRRPQGASIPPNWHIPSIQDAGTGSSIEFARL-nh2; SpyT-gDgB:Ac-GKPIPNPLLGLDSTGASAHIVMVDAYKPTKG-C(nh2)-mcpa-GASIPPNWHIPSIQDAGTGSSIEFARL-nh2

Synthetic peptide unit sequences for data in Figure 2.

SpyT-FCB-1746-gDgB:Ac-GAHIVMVDAYKPTKGSGTGGSTGRKKRSV(Citrulline)GC(nh2)-mcpa-PLKPKKPKTQEKKKKQPPKPKKPKTQEKKKKQPPKPKRGASIPPNWHIPSIQDAGTGSSIEFARL-nh2

SpyT-FCB-TAT-gDgB:Ac-GAHIVMVDAYKPTKGSGTGGSTGRKKRSV(Citrulline)GC(nh2)-mcpa-GRKRRQRRRPQGASIPPNWHIPSIQDAGTGSSIEFARL-nh2

SpyT-FCB-gDgB:Ac-GAHIVMVDAYKPTKGSGTGGSTGRKKRSV(Citrulline)GC(nh2)-mcpa-

IPPNWHIPSIQDAGTGSSIEFARL-nh2

Synthetic peptide unit sequences for data in Figure 3, Figure 4 and Figure 5.

1746-gDgB:Ac-PLKPKKPKTQEKKKKQPPKPKKPKTQEKKKKQPPKPKRGASIPPNWHIPSIQDAGTGSSIEFARL-nh2.

TAT-gDgB:Ac-GRKRRQRRRPQGASIPPNWHIPSIQDAGTGSSIEFARL-nh2.

gDgB:Ac-IPPNWHIPSIQDAGTGSSIEFARL-nh2

### 2.3. Protein Expression

The recombinant protein targeting component of the peptide vaccine (His-muXCL1-SpyC) was expressed as a fusion protein comprising an N-terminal hexa-His sequence (HHHHHH), a HRV-3C cleavage site (LEVLFQGP), murine XCL1 (residues 22-114, Uniprot P47993, Lymphotactin), a 10-amino acid linker sequence (GGSGTGATSG), followed by the SpyCatcher ligation scaffold [46] and a C-terminal Strep-II Tag (WSHPQFEK). The full-length fusion was synthesized as double stranded DNA and cloned into the pET28a+ expression vector by ATUM (Newark, CA, USA). The protein was prepared by transforming the expression vector into an *E. coli* expression strain (BL21(DE3)) and expressed in 2YT media for 16 h at 30 ·C. Expression cultures were harvested by centrifugation at 6000 g for 30 min at 4 °C, and cell pellets stored at −80 °C until purification. The protein was isolated by resuspending the cell pellets in lysis buffer followed by sonication and purification by immobilized metal affinity chromatography, buffer exchange, ion exchange chromatography and size exclusion chromatography. Final protein yield from 10 L preparation was 11.9 mg at 99% purity (University of Queensland Protein Expression Facility, PEF).

His-muXCL1-SpyC Protein sequence (24 kiloDaltons):

MGHHHHHHGATLEVLFQGPGGSVGSEVSDKRTCVSLTTQRLPVSRIKTYTITEGSLRAVIFITKRGLKVCADPQATWVRDVVRSMDRKSNTRNNMIQTKPTGTQQSTNTAVTLTGGSGTGATSGDSATHIKFSKRDEDGKELAGATMELRDSSGKTISTWISDGQVKDFYLYPGKYTFVETAAPDGYEVATAITFTVNEQGQVTVNGKATKGGAGSWSHPQFEKG

### 2.4. Vaccine Preparation

The peptide fragments containing SpyTag sequences were solubilized in sterile PBS (pH 7.4) under sterile conditions in a biohazard hood. The solubilized peptides were then incubated with the SpyCatcher-XCL1 protein in sterile PBS (pH 7.4) overnight at 4 °C. Conjugation efficiencies were determined by SDS-PAGE gel, and were 70–90%. Ligation mixtures were used directly for animal studies.

### 2.5. Mice

C57BL/6 female mice that express the CD45.2 allele were purchased from the Animal Resources Centre (Murdoch, WA, Australia). gB-specific T-cell receptor (TCR) transgenic (gBT.I) mice and gD-specific TCR transgenic mice [47] that express the CD45.1 allele were bred at the Telethon Kids Institute. Animals were housed under pathogen-free conditions. All studies were approved by the Institute’s Animal Ethics Committee (application: AE240) and conformed to the National Health and Medical Research Council Australia code of practice for the care and use of animals for scientific purposes.

### 2.6. Cell Lines and Culture

B16.F10 (B16) murine melanoma cells were purchased from the ATCC and engineered to express the HSV glycoprotein D epitope gD_290–302_ and the HSV glycoprotein B epitope gB_498–505_ (B16.gD.gB). Briefly, retroviruses were generated by transfecting the 293T cell line with pMIG-gD.gB, pMD.old.gag.pol, and pCAG-VSVG. B16 cells were transduced with 1 mL filtered retroviral supernatant in the presence of 8 μg/mL polybrene (Sigma-Aldrich, St. Louis, MO, USA). B16.gD.gB cells were passaged routinely at 70–80% confluency and cultured in RPMI media (Life Technologies, Carlsbad, CA, USA) supplemented with 10% FCS (Sigma-Aldrich), 2 mM L-glutamine, 50 µM 2-mercaptoethanol, 100 µg/mL streptomycin and 100 U/mL penicillin (all Life Technologies) at 37 °C, 5% CO_2_.

### 2.7. Prime-Boost Vaccination and Tumor Challenge

Mice were vaccinated subcutaneously on the right hind flank with either 2.0 nmol, 0.20 nmol or 0.02 nmol of vaccine constructs, in sterile PBS: 1746-gDgB, TAT-gDgB, or gDgB alone, plus 50 ug adjuvant poly I:C (Invivogen, San Diego, CA, USA). Control mice received poly I:C alone or PBS. Fourteen days later mice received a boost vaccination consisting of the same. Seven days later mice received subcutaneous inoculation of 5 × 10^5^ B16.gD.gB tumor cells, washed and injected in 100 uL sterile PBS. Tumor volume (mm^3^) was calculated using electronic calipers.

### 2.8. T Cell Expansion Assay

Single cell suspensions of transgenic T cells were prepared from pooled lymph nodes from naïve female gBT.I or gDT.II mice. After negative selection with monoclonal antibodies and magnetic beads, purity of T cells were determined by flow cytometry. 1 × 10^5^ Vα2+ CD8+ gBT.I T cells and 1 × 10^4^ Vα3.2+ CD4+ gDT.II T cells were washed and resuspended in 200 µL RPMI for i.v. injection into recipient mice at least one day prior to whole-cell vaccination. Expansion of T cells was quantified in the spleen of vaccinated mice 7 days after last challenge as the % of Vα3.2+ cells of total CD45.1+CD3+CD4+ or % Vα2+ cells of total CD45.1+CD3+CD8+ cells from the live cell gate.

### 2.9. Antibodies and Flow Cytometry

Spleen and/or lymph nodes were harvested and passed through a 70 µm metal mesh prior to red blood cell lysis. Resulting single cell suspensions were stained with monoclonal antibodies specific for mouse CD8α (53-6.7), CD45.1 (A20), Vα2 (B20.1), Vα3.2 (RR3-16) and CD4 (GK1.5) (all from BD Biosciences). Multi-parameter analysis was performed on a LSRFortessa (BD) and data were analyzed with FlowJo (Tree Star). Prior to acquisition cells were stained with propidium iodide (PI; Sigma) to exclude dead cells.

### 2.10. HSV Infection Model

Mice were anesthetized by i.p. injection of a 1:1 Ketamine (Parnell Laboratories) and Ilium Xylazil-20 (Troy Laboratories) solution in saline and HSV infections were carried out as described previously [48]. Briefly, the flank of each mouse was clipped and depilated and a small area of skin, near the tip of the spleen, was abraded. A 10 uL volume of virus, containing 10^6^ PFU, was placed onto the abraded skin and covered with OpSite Flexigrid (Smith & Nephew). The flank of the mouse was wrapped with Micropore tape and then Transpore tape (3M Health Care) to prevent removal of OpSite Flexigrid and subsequent disruption of the viral infection and wrapping was removed 48 h later.

### 2.11. Plaque Assay

A 1-cm^2^ piece of skin encompassing the infection site was removed from euthanized mice and snap frozen. The presence of infectious virus in tissue samples was determined using standard PFU assays on confluent Vero cell monolayers. Thawed skin samples were diced in DMEM media (Life Technologies) and homogenized with a TissueRuptor (Qiagen) before collecting the viral lysate. A 10-fold serial dilution series of samples were tested for plaque formation to determine viral titer in the original tissue sample.

### 2.12. Statistics

All statistical analyses were performed using GraphPad (Graphpad Software Inc., v7.0a, San Diego, CA, USA). Data are reported as a mean +/− standard error of mean for experiments comparing responses in individual animals. Differences in T cell proliferation were compared between groups using a Student’s t-test. Differences in survival were compared using the Log-Rank (Mantel–Cox) test. Statistical significance was indicated as * *p* < 0.05, ** *p* < 0.01, *** *p* < 0.001.

## 3. Results

### 3.1. Peptide Vaccine Construct without Endosomal Release Motifs Induces CD4^+^ but Not CD8^+^ T Cell Expansion

To test the approach of targeting both the vaccine construct to XCR1^+^ DC and enhancing endosomal escape of the antigen payload we designed a novel vaccine construct consisting of the murine XCL1 protein linked by SpyCatcher/SpyTag conjugation and disulphide chemistry to a CPP sequence known as 1746 [13] and the peptide sequences for HSV gB_498–505_ (gB) and gD_290–302_ (gD) [47] (Figure 1a). For comparison, we created a similar construct using a canonical CPP sequence derived from the HIV-1 TAT protein, which efficiently crosses membranes but possesses limited endosomal escape qualities. We then developed a prime-boost vaccination strategy incorporating Poly(I:C) as the vaccine adjuvant, where we first seed a low precursor frequency of antigen-specific CD8^+^ (gBT.I) and CD4^+^ (gDT.II) [47] T cells into naïve C57Bl6 mice prior to vaccination and subsequent boosting 2 weeks later. One week after the boost vaccination we harvested spleens from mice to analyse the expansion of antigen-specific T cells in response to vaccination or relevant controls (Figure 1b). Using this experimental design, we found that both 1746-gDgB and TAT-gDgB vaccines conjugated to XCL1 produced similar expansion of gDT.II T cells. However, this expansion was not greater than that observed for a control vaccine containing no CPP suggesting, as expected, that the CPP is not required for antigen presentation on MHC class II (Figure 1c). Surprisingly, we saw little to no expansion of gBT.I T cells in this system leading us to believe that something in our construct design was inhibiting the export of antigen from the endosome and thus preventing the cross-presentation of vaccine antigens.

### 3.2. Addition of Furin and Cathepsin B Cleavage Motifs Facilitates Endosomal Cleavage and Cross-Presentation of Vaccine Antigens

We hypothesized that limited reduction of the disulphide bond within the vaccine construct was potentially occurring, leading to improper release of the antigenic subunit from the SpyTag and thus limiting endosomal escape and entry into the cross-presentation pathway. Disulphide bond reduction is feasible within the endosomal uptake pathway [49,50], despite reduction being less likely in the low pH environment of endosomes. However, it has also been reported that recycling and late endosomes specifically are not reducing environments, but in fact are oxidising [50]. Therefore, we engineered a cleavable furin/cathepsin B (FCB) motif into the vaccine construct (Figure 2a) to facilitate the release of the CPP-antigen payload from the larger SpyCatcher-SpyTag complex. Indeed, when we repeated the same prime-boost vaccination strategy with these new vaccine constructs and analyzed T cell expansion we saw robust gDT.II (Figure 2b) and gBT.I (Figure 2c) T cell expansion in response to vaccination. Encouragingly, we observed that the 1746-gDgB vaccine enhanced antigen-specific gBT.I CD8^+^ T cell expansion by greater than 2-fold over TAT-gDgB, which showed no improvement over the no-CPP vaccine. Importantly, these data suggest that addition of the FCB cleavage motif allows for proper decoupling of vaccine within endosomes, whereafter the antigenic component is carried into the cytoplasm through the endosomal escape properties of the 1746 CPP, linked to the peptide antigens, facilitating enhanced cross-presentation.

### 3.3. Non-Targeted Peptide Vaccine, Optimized for Endosomal Escape, Induces Robust CD8^+^ T Cell Expansion at Low Concentrations

From the results of the previous two experiments it seemed that vaccine targeting to XCR1^+^ cross-presenting DC alone was likely to be insufficient to drive robust cross-presentation, if the antigens were restricted within endosomes. Therefore, we next tested with non-targeted CPP-vaccine constructs, as this reduced the need to uncouple vaccine components within intracellular compartments. Again we used our prime-boost vaccination strategy with 1746-gBgD, TAT-gDgB or gDgB peptide vaccines and measured the expansion of gDT.II and gBT.I T cells at 7 days post boost vaccination. Here we saw that gDT.II expansion was similar between 1746-gBgD and TAT-gDgB, with both showing improvement over the no-CPP vaccine (Figure 3a). Critically, in the absence of XCR1-targeting, the advantage of the 1746 CPP over TAT was retained, with a significant improvement in gBT.I T cell expansion measured with 1746-gDgB in this setting, while TAT-gDgB did not significantly improve expansion over the no-CPP vaccine (Figure 3b).

We then probed the efficiency of our vaccine approach. Vaccinating with a titration of 2.0 nM, 0.2 nM or 0.02 nM of vaccine constructs and again assaying antigen-specific T cell expansion in the spleen of mice 7 days after the boost vaccination. Interestingly, we saw that 1746-gDgB maintained both gDT.II (Figure 3c) and gBT.I T cell (Figure 3d) expansion at a similar level when we decreased the vaccine dose by 10-fold, while TAT.gDgB did not have this effect. Indeed, the 1746.gDgB vaccine, drove more expansion of gBT.I cells at 0.2 nM than the TAT.gDgB vaccine did at 2 nM, a 10-fold higher dose, suggesting that the endosomal escape capability of the 1746 CPP is incredibly efficient at driving antigen out of endosomes for cross-presentation. These data also suggest that vaccine antigens are acquired efficiently enough by DC, without the need for targeting, so long as those antigens can efficiently access the cross-presentation machinery within the DC.

### 3.4. Prophylactic Non-Targeted Vaccine Reduces Herpes Simplex Virus 1 Viral Tire and Zosteriform Skin Lesion Severity at the Site of Infection

Having determined the effectiveness of our CPP coupled vaccine to drive vaccine-specific T cell expansion, we now wanted to see whether the CD8^+^ T cell response that was generated was functionally able to control a HSV skin infection, where the virus natively expresses the vaccine antigens. We vaccinated mice with a single dose of vaccine in combination with Poly(I:C) one week prior to performing a skin scarification and HSV infection. At a timepoint where viral recrudescence is visible as a result of viral replication in sensory neurons (D6 post infection), we harvested skin to perform viral plaque forming assays to titrate viral load at the site of infection (Figure 4a). We found that vaccination with 1746-gBgD vaccine reduced viral titres compared to TAT-gBgD, which again were similar to no-CPP peptide vaccine controls, suggesting that endosomal escape of antigen is driving expansion of functional virus-specific T cells that control the infection. Indeed, in mice vaccinated with 1746-gBgD, we found that 7/9 (78%) mice had no detectable virus at the site of infection, compared to 5/9 (55%) mice for TAT-gBgD (Figure 4b), and those that did have detectable virus had visibly reduced herpes zosteriform skin lesions (Figure 4c). All peptide vaccines tested reduced viral titres and visual signs of infection compared to Poly(I:C) only controls, however the 1746-gDgB vaccine clearly out-performed the other constructs. These data demonstrate that the vaccine-specific T cell repertoire generated by 1746-gDgB vaccine is functionally able to control viral infection and clear virus from the site of infection, however, this does not exclude a role for enhanced humoral immunity in this setting. Such viral models may prove to be an efficient method to quickly screen new candidate vaccine constructs before testing in more complex and time-consuming preclinical cancer models, where neoantigens are often less well characterized.

### 3.5. Prophylactic Non-Targeted Vaccine Reduces Tumor Growth and Prolongs Survival in Mice with B16 Melanoma, Expressing gD and gB Antigens

Given the promising results for our endosomal escaping CPP-vaccine construct in our T cell expansion assays and HSV infection model we wanted to determine whether this strategy could generate potent anti-tumor immunity. We used a B16 melanoma model, engineered to express the vaccine epitopes gD and gB and applied the same prime boost regimen used previously in our expansion assays. The simple gDgB peptide vaccine increased overall survival by approximately 2 weeks, a non-significant increase compared to the adjuvant alone. Vaccinating with 1746-gDgB extended survival by a further two weeks (Figure 5a), achieving a significant increase over the adjuvant only controls, but not the gDgB vaccine. Tumors arose more slowly in vaccinated animals, suggesting immune surveillance was increased in this setting (Figure 5b). In fact, tumors in 1746-gBgD vaccinated mice grew similarly to those given a prior HSV infection, which acts as a live virus vaccine, containing the same antigens expressed on the tumor cells (Figure 5b). These data suggest that vaccination with an endosomal escape CPP-linked vaccine construct is a viable strategy to prime functional anti-tumor immunity to tumor expressed neoantigens. Our data, from T cell expansion assays, suggests this process is more efficient when the vaccine construct is linked to a CPP and although vaccination alone failed to control tumor outgrowth and increase survival this strategy may synergise well with other forms of immunotherapy such as checkpoint blockade to provide improved clinical outcomes.

## 4. Discussion

Here we have reported a novel strategy to increase the efficiency of peptide-based cancer vaccines, using a dual approach of targeting the vaccine construct to XCR1^+^ cDC1 and enhancing antigen entry into the cross-presentation pathway through the use of an endosomal escape CPP. We find that the latter approach is key for enhanced antigen specific CD8^+^ T cell expansion following vaccination and that correct endosomal cleavage and subsequent escape is critical for cross-priming to occur.

DC-targeting as a route to enhance the potency of cancer vaccines has been explored previously, with targeting of antigen to DC-specific markers such as XCR1 [33,34], Clec9A [51,52] and DEC205 [53,54]. It is well established that uptake of antigen associated with these receptors can favor cross-presentation [55,56]. However, the exact mechanisms remain to be fully elucidated. Our results with both targeted and non-targeted vaccine constructs show robust CD8^+^ T cell expansion, suggesting that targeting, in this setting, is not an absolute requirement. This may be explained by the fact that the skin is a relatively DC-rich organ that undergoes constant immune surveillance [57] and, as a result, uptake of the vaccine by DC occurs efficiently without the need for targeting.

Rather, the key to success in our targeted vaccine design appears to be engineering of the construct to allow decoupling of the targeting and CPP-antigen subunits. Our initial hypothesis that a disulfide linked peptide vaccine construct would be reduced in the endosome to facilitate vaccine decoupling was shown to be flawed (Figure 1). While it has been reported that disulfide reduction can occur with endosomal uptake and processing [49] another study investigating antibody-drug conjugates found, on the contrary, that endosomes and lysosomes are oxidizing environments [50]. When we altered the vaccine construct to include surface and endosome protease, Furin and late-stage endosome protease, Cathepsin-B [58] cleavage motifs, there was a marked increase in antigen-specific CD8^+^ T cell expansion (Figure 2). Therefore, we conclude that these cleavage motifs facilitate proteolytic uncoupling of the targeted vaccine construct and enable the 1746 CPP to drive endosomal escape of antigenic epitopes for loading into the cross-presentation pathway.

The use of CPP as a strategy to enhance vaccine-delivered antigen uptake and cross-presentation in DC is an area of growing research. Originating with sequences identified from nature such as the HIV-derived TAT peptide [59] and Penetratin [60], which can shuttle cargos across cell membranes, modern CPPs are now engineered to enhance their cell-penetrating and endosomal escape capabilities [13,61] or to avoid entrapment within endosomes altogether [62]. This may account for earlier reports suggesting that CPP-linked antigens were no more effective than those undergoing receptor mediated endocytosis for targeting DC for cross-presentation [63] and our own data, comparing TAT to the 1746 CPP support this. More recently, preclinical studies have demonstrated the effectiveness of CPP-linked vaccines in prophylactic vaccination against OVA-expressing EG7 tumors [64], as well as metastatic melanoma [65], combining CPP, antigen and adjuvant in unique vaccine constructs allowing for the spatial-temporal integration of antigen uptake and directed APC activation [66]. Indeed, while the mechanisms of CPP mediated entry into cells, and specifically DC, have been well documented [67,68,69], the more critical processes governing endosomal escape and access to the cross-presentation pathway remain less understood [70,71].

Clinical trials using novel CPP containing vaccines such as SO-11 KISIMA-01, a Phase Ib trial in stage IV colorectal cancer [72] utilising AMAL Therapeutics KISIMA platform, in combination with the PD-1 antibody enzabenlimab, are already underway. In combination with the recent progress made in neoantigen discovery pipelines [73,74], where CPPs may also be useful in the rapid screening of candidate neoantigens, the clinical application of adding CPP to cancer vaccines appears promising. Many companies are now developing intellectual property in this area, however, despite this rapid increase in development, no CPP-based therapeutic with an indication in oncology is currently FDA approved [75]. Importantly, biological diversity represents a rich source of CPPs for future development. Indeed, Bolhassani et al., recently identified numerous potential CPP sequences derived from the proteasome of SARS-CoV-2 [76] and the repurposing of viral CPPs to enhance the delivery of mRNA vaccines is currently being investigated [77], demonstrating the wide-reaching utility of CPP across a broad-spectrum of future immunological applications.

## 5. Conclusions

Here we demonstrated that the addition of an endosomal escape CPP to a simple peptide vaccine construct increases the efficiency of vaccination, driving strong antigen specific CD8^+^ T cell expansion. A key requirement of this approach was the correct decoupling of targeting moieties from the CPP-antigen subunit within the endosome, allowing for endosomal escape and entry into the cytosolic cross-presentation pathway. Our CPP-vaccine construct primed immunity that reduced viral load after HSV skin infection and displayed an anti-tumor effect in a preclinical melanoma model, engineered to express the vaccine antigens. These results provide preclinical proof of concept for the use of endosomal escaping CPP for the development of effective cancer vaccines.

## 6. Patents

Data in this manuscript have been published as part of provisional patent application WO/2019/134018: Vaccine conjugates and uses thereof. Jason Waithman, Shane Stone, Paul Watt.

## Figures and Tables

**Figure 1 cancers-13-06189-f001:**
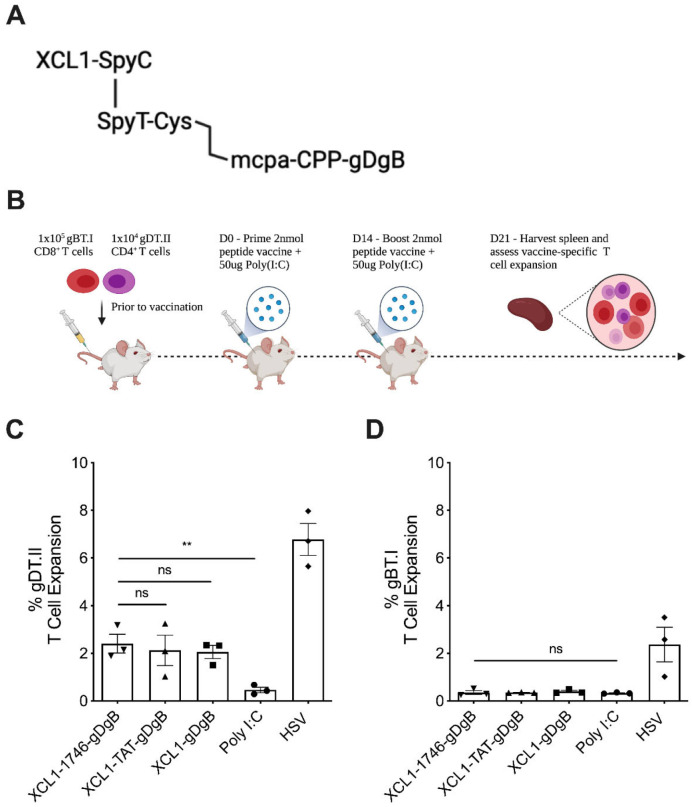
T cell expansion following prime-boost vaccination with XCL1-targeted, CPP-vaccines: (**A**) The peptide vaccine construct comprises an XCR1 targeting unit (XCL1-SpyC) linked to a vaccine unit (SpyT-Cys-Cys-CPP-gBgD). The disulphide linkage (Cys-Cys) should dissociate to release the CPP-gDgB subunit within the endosome. (**B**) Treatment schedule; mice received adoptive transfer of 1 × 10^5^ gBT.I and 1 × 10^4^ gDT.II T cells prior to subcutaneous prime and boost vaccinations with 2 nM of vaccine constructs plus 50 ug Poly(I:C). (**C**) gDT.II (CD4^+^) and (**D**) gBT.I (CD8^+^) T cell expansion was measured as percentage of the total CD4^+^ or CD8^+^ pool 7 days after the boost vaccination. Data are shown for the different vaccine constructs (XCL1-1746-gDgB, XCL1-TAT-gDgB or XCL1-gDgB) plus Poly(I:C) alone as negative control. HSV infection (natively expressing vaccine epitopes) was used as a positive control. Data are from a single experiment and *n* = 3 three mice per group, displayed as mean +/− SEM. ** *p* < 0.01, ns = not significant. Image created with Biorender.com, under license.

**Figure 2 cancers-13-06189-f002:**
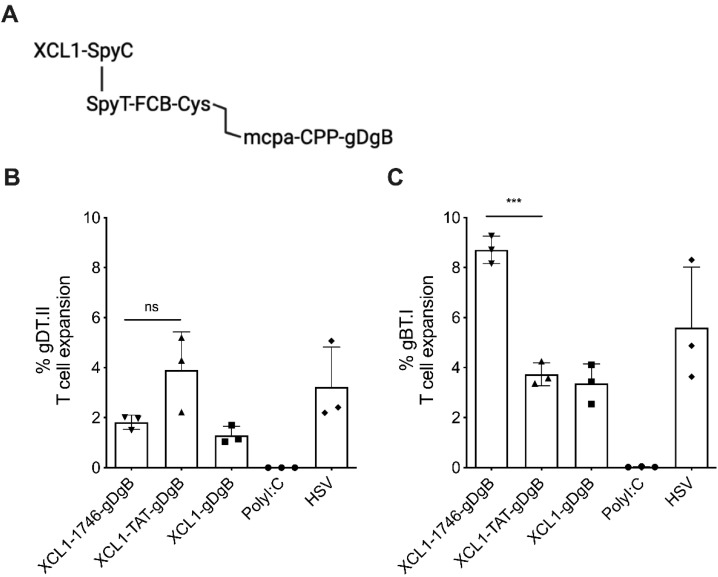
Addition of furin/cathepsin B cleavage motifs to vaccine constructs enhances gBT.I CD8^+^ T cell expansion following prime-boost vaccination with XCL1-targeted 1746-vaccine: (**A**) The peptide vaccine construct comprises an XCR1 targeting unit (XCL1-SpyC) linked to a vaccine unit (SpyT-FCB-Cys-Cys-CPP-gBgD) containing furin/cathepsin B cleavage motifs (FCB) to facilitate uncoupling of the vaccine within the endosome to allow for endosomal escape. Treatment schedule; mice received adoptive transfer of 1 × 10^5^ gBT.I and 1 × 10^4^ gDT.II T cells prior to subcutaneous prime and boost vaccinations with 2 nM of vaccine constructs plus 50 ug Poly(I:C). (**B**) gDT.II (CD4^+^) and (**C**) gBT.I (CD8^+^) T cell expansion was measured as percentage of the total CD4^+^ or CD8^+^ pool 7 days after the boost vaccination. Data are shown for the different vaccine constructs (XCL1-1746-gDgB, XCL1-TAT-gDgB or XCL1-gDgB) plus Poly(I:C) alone as negative control. HSV infection (natively expressing vaccine epitopes) was used as a positive control. Data are from a single experiment, representative of two similar experiments, and *n* = 3 three mice per group, displayed as mean +/− SEM. *** *p* < 0.001, ns = not significant.

**Figure 3 cancers-13-06189-f003:**
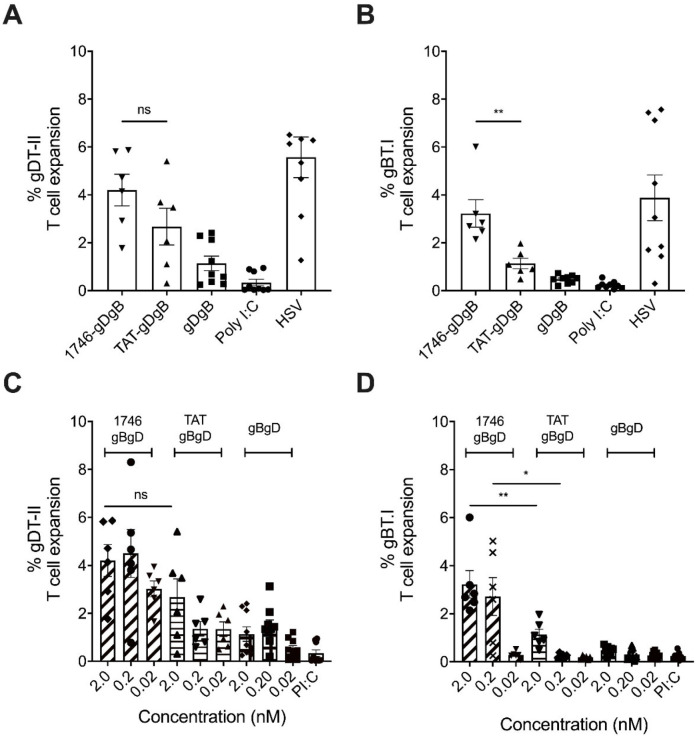
Non-targeted 1746-gBgD vaccine capable of endosomal escape drives robust expansion of gBT.I CD8^+^ T cells at 10-fold lower concentration than TAT-gBgD vaccine. Treatment schedule; mice received adoptive transfer of 1 × 10^5^ gBT.I and 1 × 10^4^ gDT.II T cells prior to subcutaneous prime and boost vaccinations with 2 nM, 0.2 nM or 0.02 nM of vaccine constructs plus 50 ug Poly(I:C). (**A**) gDT.II and (**B**) gBT.I T cell expansion after vaccination with 2 nM of non-targeted vaccine constructs was measured as percentage of the total CD4^+^ or CD8^+^ pool 7 days after the boost vaccination. Data are from two or three independent experiments and *n* = 6 or *n* = 9 for each group, displayed as mean +/− SEM. (**C**) gDT.II and (**D**) gBT.I T cell expansion after vaccination with 2 nM, 0.2 nM or 0.02 nM of untargeted vaccine constructs was measured as percentage of the total CD4^+^ or CD8^+^ pool 7 days after the boost vaccination. Data are from two or three independent experiments and *n* = 6 or *n* = 9 for each group, displayed as mean +/− SEM. ** *p* < 0.01, * *p* < 0.05, ns = not significant. Diagonal bars = 1746-gDgB, Vertical bars = TAT-gDgB, Squares = gDgB.

**Figure 4 cancers-13-06189-f004:**
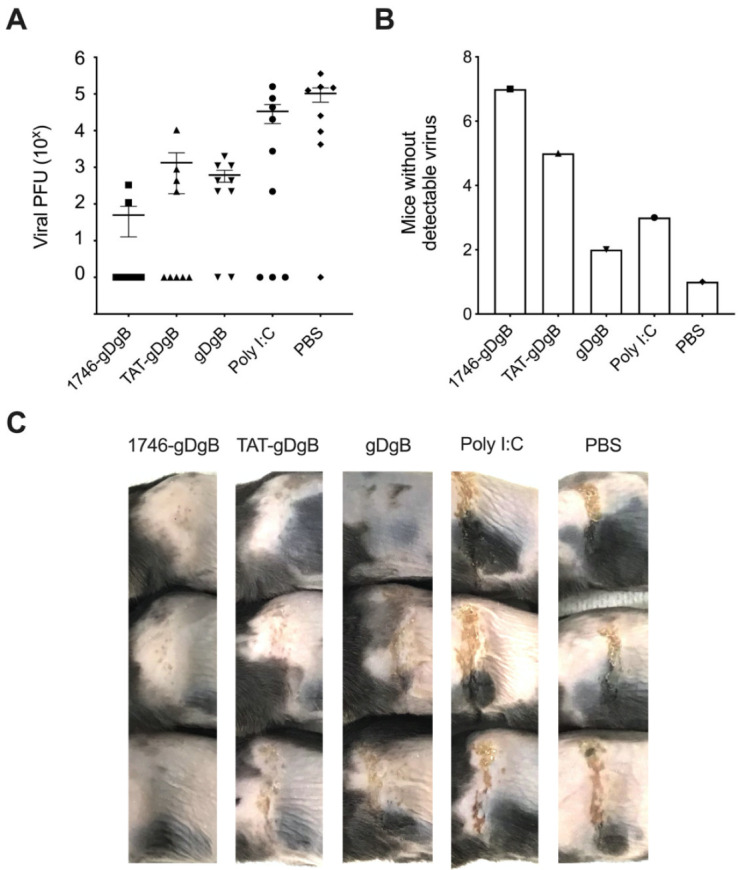
The 1746.gDgB vaccine generates a functional antigen-specific T cell repertoire capable of reducing HSV-1 skin infection. Treatment schedule; mice received subcutaneous vaccinations with 2 nM of vaccine constructs plus 50 ug Poly(I:C), one week prior to HSV skin infection, on the opposite flank. No prior transfer of transgenic T cells were given before infection. (**A**) Skin was taken from the infected site at the peak of viral infection (D6) and a viral plaque forming assay was performed to titrate the amount of virus present in the skin. (**B**) The number of mice from each group which had no detectable viral load in their skin sample. (**C**) Photographs depict the skin infection site of three representative mice from each treatment group. Herpes zosteriform skin lesion is clearly visible in Poly(I:C) and PBS treated animals and reduced in vaccine groups. Data are from three independent experiments and *n* = 9 for each group, displayed as mean +/− SEM. Three animals from a single representative experiment were used for photographs.

**Figure 5 cancers-13-06189-f005:**
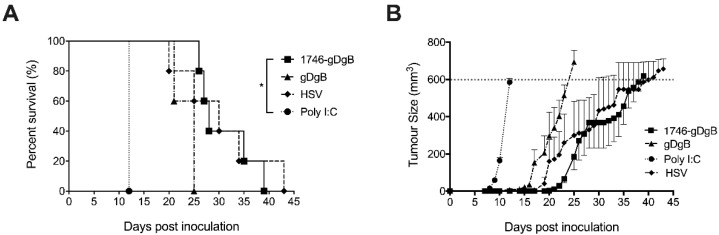
Prophylactic vaccination with a 1746 CPP-linked vaccine construct delays tumor outgrowth and prolongs survival. Treatment schedule: Mice received subcutaneous prime and boost vaccinations with 2 nM of vaccine constructs plus 50 ug Poly(I:C), or HSV infection. One week later, on the opposite flank, mice were subcutaneously inoculated with B16 melanoma cells expressing the HSV-derived antigens gB and gD, contained within the vaccine constructs. No prior transfer of transgenic T cells was given before tumor inoculation. (**A**) Survival plots of mice that received 1746-gBgD vaccine, no-CPP gBgD vaccine, HSV infection or Poly(I:C) alone. (**B**) Growth curves for tumors measured in mm3. Data are from a single experiment and *n* = 5 mice per group, displayed as mean +/− SEM. * *p* < 0.05: Log-rank (Mantel-Cox) test.

## Data Availability

The data that support the findings of this study are available from the corresponding author upon reasonable request.

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
