# Peer review of "Targeting Cross-Presentation as a Route to Improve the Efficiency of Peptide-Based Cancer Vaccines"

_cancers, 2021, doi:10.3390/cancers13246189_

Round 1

Reviewer 1 Report

The authors addressed all concerns

Reviewer 2 Report

The authors addressed all previous comments in the current revised version. 

This manuscript is a resubmission of an earlier submission. The following is a list of the peer review reports and author responses from that submission.

Round 1

Reviewer 1 Report

The manuscript "Targeting cross-presentation as a route to improve the efficiency of peptide-based cancer vaccines" presents an improved peptide-based cancer vaccine by adding an XCR1-targeting moiety to target DCs and CPP to escape endosomes. The manuscript is well-organised, and the results clearly proved the authors' hypothesis. Although the reviewer's personal desire wants to see the data presenting the role of XCR1 and CPP in the cellular levels - such as more vaccine materials in the presence of XCR1-targeting moiety and presence of the vaccines in the cytoplasm - in the manuscript, the current version of the manuscript is still good enough to be accepted. 

Author Response

We thank the reviewer for their assessment of our manuscript. We appreciate the need to elucidate the cellular mechanisms underlying improved vaccine efficiency in our future work.  

Reviewer 2 Report

The authors provide interesting data showing that peptide vaccines integrating a cell penetrating peptide (CPP) including a furin and cathepsin B motif promote cross presentation and robust stimulation of CD8 T cell responses in addition to CD4 T cell responses. The authors provide proof-of-concept using a HSV model and preliminary data using the B16 mouse melanoma model. While promising, several issues require further clarification:

  1. In section 2.1 the authors introduce HSV glycoprotein B (gB) epitope as gB495-505, suggesting this is an 11-mer peptide. However, section 2.2 lists this peptide as an 8-mer. Also, section 2.8 says 5x10E4 gBT.I cells were adoptively transferred per mouse, but figure legends say 1x10E5.
  2. The ratio of gBT.1 (CD8) to gDT.II (CD4) cells used for adoptive transfer was 10:1. This suggests that relatively speaking cross-presentation was not very effective. The authors should further discuss the implications of this.
  3. I assume the HSV infection and B16 tumor challenge experiments did not include adoptive transfer of transgenic T cells, but the authors might want to clarify that.
  4. Can the authors exclude the possibility that anti-HSV antibodies were (differentially) induced by the peptide vaccines, and contributed to control of HSV?
  5. It is unclear why the authors did not use a B16 peptide vaccine based on earlier validated tumor antigens in the B16 model. By cloning HSV antigens into B16, the model is arguably a poor surrogate for tumor antigens. Targeting B16 tumor antigens would also have permitted benchmarking the authors’ vaccine strategy with other cancer vaccine strategies.
  6. Unlike the HSV-related studies, the studies performed in the B16 model are preliminary without any mechanistic studies. As such, it seems a stretch to claim that the modified peptide vaccines improve cancer vaccines, as the title and other parts of the manuscript currently do.

Author Response

We thank our reviewers for their insightful suggestions and greatly appreciate their overall very positive appraisal of our study. We have modified our manuscript according to their queries and comments. Our specific point-by-point reply is outlined below with changes to the manuscript file highlighted by track changes throughout.

  1. The reviewer is correct. The gB epitope is 498-505 not 495-505 and this was a conserved typo in the text, this change along with the change in cell number in section 2.8 to 1x10^5 gBT.I cells has now been made in the amended manuscript.
  2. The specific number of CD8+ and CD4+ T cells was chosen to mimic T cell precursor frequencies in naive mice as published previously (Gebhardt T, et al., Nature 2011). As such antigen-specific T cell expansion is measured as a percentage of the entire CD4 or CD8 T cell pool.

  3. The reviewer is correct. Experiments examining the prophylactic effect of vaccination against either HSV virus or B16 tumours did not include adoptive T cell transfer. The line “No prior transfer of transgenic T cells was given” has been added to the figure legend of each relevant figure to clarify this.

  4. The reviewer raises a good point. In our experiments we focused specifically on the ability of the vaccine construct to enhance CD8 and CD4 T cell expansion and the resultant effects on viral and tumour control in a similar fashion to Mackay et al., PNAS, 2012. We have included a line in the text to state that our study “does not exclude a role for enhanced humoral immunity”.

  5. The authors make a valid point. Current cancer vaccination strategies often employ peptide pools consisting of 10-20 target peptides as in Sahin et al., Nature, 2017. For this project we utilised a model antigen approach, to allow for rapid screening, development and validation of our vaccination strategy. The HSV antigen model is already well characterised and reported in the literature (Mueller et al., JEM, 2002; Gebhardt et al., Nature, 2011; Park et al., Nature, 2018) and therefore was appropriate for this application.  Future studies would indeed benefit from using published antigens from the B16 melanoma model, as the question of peptide antigen selection is of significant relevance to such work.

  6. We accept the reviewer’s comments on preliminary nature of the data. We note that the title only refers to ‘targeting cross-presentation is a route to improve efficiency’ and not anti-cancer efficacy of the cancer vaccine. To not overstate the conclusions drawn from the data we have edited some of the wording in the text, relating to whether or not an improved effect was observed between the no-CPP and CPP-vaccine constructs.

Reviewer 3 Report

In this manuscript, the authors present a novel approach for the development of therapeutic peptide –based vaccines specifically targeting cross-presentation pathways. In particular, the authors claim that the introduction of protease cleavage motifs in the peptide vaccine formulation increases endosomal escape, favoring antigen cross-presentation and ultimately leading to improved vaccine efficacy. While the rationale of the study is logical, scientifically sound and well explained, in some instances data are insufficient or do not necessarily support all the reported conclusions:

  • In the first lines of the introduction it is mentioned that ACT showed good results in blood cancers, this is not completely true, as the references also mentioned in the text ACT showed promising therapeutic benefits in solid tumors such as melanoma. This sentence should be reformulated.
  • In section 2. GFP+ cells are mentioned, although it is not clear what the authors refer to, since GFP constructs are not mentioned anywhere else in the text.
  • Figure 1 and 2 report data from 1 single experiment and three replicates. I believe this is not sufficient and the reported observations should be repeated in additional experimental replicates, as correctly done for subsequent figures (e.g. Figure 3) to verify data consistency.
  • Figure 2: the vaccine nomenclature used in this figure is exactly the same as the one reported in Figure 1, although the vaccine constructs are slightly different (i.e. insertion of cleavage motifs). To avoid confusion, the nomenclature should be here changed to reflect these introduced differences.
  • Figure 5 and paragraph 3.5: the authors reported a statistical significance in mouse survival only between their optimal vaccine construct (1746-gDgB) which includes a cleavage motif improving cross-presentation, compared to the adjuvant control. However, no statistical significance was reported between this vaccine construct and the canonical construct with no cleavage motif (gDgB). If this is confirmed, this observation does not support the claim that the vaccine platform here presented with the introduction of cleavage motifs produces any therapeutic benefits/improvements over canonical peptide vaccine formulations (i.e.: the sentence: “Our data suggest this process is more efficient than standard peptide vaccination” is not supported by the presented evidence). Furthermore, although the experimental setup here used is sound and largely used, to accurately verify therapeutic vaccination benefits, a model in which the tumor is implanted before vaccination and animals are subsequently vaccinated (and not other way around as here presented) would better reflect a clinical therapeutic situation and should be potentially introduced as a complementary approach.

Few minor edits and typos should also be addressed:

  • Paragraph 2.7: “receive” instead of “receive”
  • Paragraph 3.2, page 10: “over the a- no CPP vaccine” should be revised
  • Paragraph 4: “dendritic cells” should be indicated with the introduced abbreviation “DC”
  • Paragraph 5: “A key requirement of this approach wa the correct”, please correct typo

Author Response

We thank our reviewers for their insightful suggestions and greatly appreciate their overall very positive appraisal of our study. We have modified our manuscript according to their queries and comments. Our specific point-by-point reply is outlined below with changes to the manuscript file highlighted by track changes throughout. Minor edits have been addressed in the text.

  1. We have adjusted the sentence to read “and adoptive cell therapy (ACT) [4-6], for the treatment of blood cancer, as well as limited solid tumors."
  2. We thank the reviewer for picking up this error, the references to GFP+ cells in the methods were not applicable for this project and have been removed.

  3. We understand the reviewers concerns with data from single experiments.

    1. For Figure 1, the data was clearly negative and further literature review and analysis identified that endosomal pH conditions in cross-presenting DC are altered by NOX2 to favour antigen retention, which had implications for our vaccine design (Savina et al., Cell, 2006). Therefore, instead of repeating negative data we went back to redesign the next iteration of our vaccine construct to include the FCB cleave motifs necessary in this setting. This constitutes a novel and important aspect of the study.

    2. For Figure 2, we performed multiple similar experiments to optimise and determine saturating vaccine dose and other experimental variables. These repeats were not all uniform in their experimental parameters and as such cannot be collated into a single figure. We have updated the figure legend to read ‘Data are from a single experiment, representative of two similar experiments’ and would like to include the reviewer only figure (attached) as justification.
  4. Figure 2A, which details the structure of the new construct contains an FCB section (SpyT-FCB-Cys), which is not present in the Figure 1A, to represent the relative position of the FCB insertion, we apologise if this is not clear. The description of the construct in the figure legend also contains: ‘vaccine unit (SpyT-FCB-Cys-Cys-CPP-gBgD) containing furin/cathepsin B cleavage motifs (FCB) to facilitate uncoupling of the vaccine within the endosome'. The nomenclature in the figures themselves has not been updated for brevity. It can be inferred that any construct used after Figure 1 contains the FCB element necessary for proper antigen cross-presentation.

  5. The reviewer’s point is valid and, as per Reviewer 2’s request, we have altered the wording of several conclusions to reflect that there was no significant survival benefit between the no-CPP vaccine and CPP-vaccine, although there was benefit over adjuvant alone for CPP-vaccine, but not for the no-CPP vaccine. The wording ‘efficient’ was chosen over ‘efficacious’ for this reason, as supported by the results for Figure 3C/D looking at vaccine dose. Further analysis of the growth curves does reveal significantly delayed tumour outgrowth at D21, 22, 23 and D25 post-injection when comparing the mean tumour size of the no-CPP vaccine and CPP-vaccine groups (p<0.05, grouped analyses, multiple t Tests, GraphPrism), however, in this case we believe that presenting the complete data set for overall survival is more representative of a real effect.
  6. The reviewer’s point is well made, effectiveness in a therapeutic setting would certainly strengthen the data. Currently, prophylactic vaccination is the gold standard in the field for testing and screening applications. The experimental set up for a therapeutic vaccination is quite different and, in the B16 melanoma model, presents quite a high bar, without other combination of immunotherapy. Therefore, for this proof of principle study, where we were focused on optimising the vaccine construct, we chose to use the prophylactic approach. 
